# Regression of Neovascularization after Panretinal Photocoagulation Combined with Anti-VEGF Injection for Proliferative Diabetic Retinopathy—A Review

**DOI:** 10.3390/diagnostics14010031

**Published:** 2023-12-22

**Authors:** Maciej Gawęcki, Krzysztof Kiciński, Lorenzo Bianco, Maurizio Battaglia Parodi

**Affiliations:** 1Dobry Wzrok Ophthalmological Clinic, 80-822 Gdansk, Poland; 2Department of Ophthalmology, Pomeranian Hospitals, 84-200 Wejherowo, Poland; 3Department of Ophthalmology, Specialist Hospital, 89-600 Chojnice, Poland; krzysztofkg999@icloud.com; 4Department of Ophthalmology, IRCCS San Raffaele Scientific Institute, 20132 Milan, Italymaubp@yahoo.it (M.B.P.)

**Keywords:** proliferative diabetic retinopathy, panretinal photocoagulation, intravitreal injection, vitreous hemorrhage, pars plana vitrectomy

## Abstract

Proliferative diabetic retinopathy (PDR) poses a significant therapeutic problem that often results in severe visual loss. Panretinal photocoagulation (PRP) has long been a mainstay treatment for this condition. Conversely, intravitreal anti-VEGF therapy has served as an alternative treatment for PDR. This review aimed to evaluate the effects of PRP combined with anti-VEGF therapy on the regression of neovascularization (NV), including functional outcomes and incidence of complications. The MEDLINE database was searched for articles evaluating regression of NV using a combination of the following terms: “proliferative diabetic retinopathy”, “anti-VEGF”, “panretinal photocoagulation”, and “combined treatment”. The search yielded a total of 22 articles. The analysis of their results indicated PRP combined with ant-VEGF therapy as superior over PRP alone in the management of PDR. Combination treatment yields better and faster regression of NV and a lower incidence of serious complications, such as vitreous hemorrhage and the need for pars plana vitrectomy. Nevertheless, complete regression of NV is not achieved in a significant proportion of patients. Further research is needed to establish the most effective schedule for intravitreal injections as an adjunct to PRP. The current literature shows that in some cases, cessation of anti-VEGF injection in combination treatment for PDR can lead to relapse of NV.

## 1. Introduction

Diabetic retinopathy is one of the leading causes of visual loss in developed countries. Among its subtypes, proliferative diabetic retinopathy (PDR) is considered a vision-threatening diabetic retinopathy, as it is burdened with severe potential complications that affect patients’ sight. According to the Wisconsin Epidemiology Study for Diabetic Retinopathy (WESDR), the prevalence of PDR is 23% in patients with an earlier onset of diabetes, 10% in those with a later onset of diabetes, and 3% in those who do not take insulin [1]. The 4-year risk of the development of PDR increases with the presence of hyperglycemia, a longer duration of diabetes, and a more severe retinopathy at baseline. In the WESDR, a cumulative 25-year risk of progression to PDR of 42% is reported [2]. This finding shows that the total number of patients with PDR is relatively large, indicating the need for close observation and frequent prompt initiation of treatment. The serious complications of PDR include vitreous hemorrhage (VH), vitreoretinal traction, and tractional retinal detachment or retinal tears that can lead to retinal detachment through a rhegmatogenous mechanism, which usually require surgical intervention, including pars plana vitrectomy (PPV). The outcomes of vitreoretinal surgery for advanced forms of PDR are often unsatisfactory. Despite achieving good morphological outcomes, patients often experience some degree of visual impairment [3,4].

Early initiation of PDR treatment can prevent serious complications and result in better visual outcomes in the long term [5,6]. Thus, the aim of modern ophthalmology is to provide therapeutic solutions that would minimize the need for surgical intervention and achieve satisfactory functional outcomes.

Panretinal photocoagulation (PRP) has long been the mainstay treatment for PDR. Since the first Diabetic Retinopathy Study Group studies in the 20th century, much quality randomized research has confirmed the efficacy and reliability of PRP [7,8,9]. However, the mechanism of retinal ablation via laser photocoagulation remains poorly understood. The most prevalent pathophysiological concept is based on the oxygen theory [10,11]. According to this concept, PRP creates scars that serve as bridges for the oxygen flow from the choroid to the inner parts of the retina. Accordingly, oxygenation of the retina significantly improves, reducing the production of vasoproliferative cytokines [12]. This effect is enhanced by the reduction of retinal need for oxygen as a consequence of destruction of a large number of photoreceptor cells that normally consume large amounts of oxygen [13]. Nonetheless, PRP is a destructive procedure associated with potential complications such as restriction of the visual field, proliferation of glial cells on the retinal surface, formation of epiretinal membranes, and detachment or neovascularization (NV) of the choroid [14]. Moreover, the procedure is not 100% effective despite extensive ablation of the peripheral retina. Some studies report that in as much as 20–30% of patients, PDR does not regress or deteriorate despite receiving full laser treatment [15,16,17].

Modern ophthalmology is directed toward retina-sparing forms of treatment, with intraocular injections of anti-VEGF agents considered to be at the first line. Anti-VEGF therapy works by blocking receptors for vasoproliferative cytokines that are elicited by ocular tissues in the state of hypoxia. Consequently, VEGF-related neo-vessel growth is hampered, and VEGF-related hyperpermeability of retinal vessels is reduced. Drugs have been originally formulated for the treatment of macular diseases, including exudative forms of age-related macular degeneration, diabetic macular edema (DME), and macular edema secondary to retinal vein occlusion [18,19]. Nevertheless, they have been administered off-label as auxiliary procedures to minimize bleeding during and after PPV, speed up the resorption of intravitreal hemorrhage, or, occasionally, enhance the regression of NV in proliferative retinopathies [20,21,22,23]. Despite the marked progress in the evolution of treatment schedules for anti-VEGF drugs, they must still be administered at a relatively high frequency to ensure effectiveness. Moreover, achieving permanent regression of the treated clinical entity only through intravitreal injections and without the need for further treatment remains controversial [24].

As both PRP and intravitreal anti-VEGF therapy target the development of NV, combining them for the treatment of PDR appears intuitive. The potentially additive actions of both therapies could theoretically yield better regression of neo-vessels and prolong the durability of the therapeutic effect without the need for extensive ablation of the peripheral retina. Lighter PRP is associated with visual field preservation and fewer complications, as proven in studies comparing between light and classic retinal photocoagulation and multispot laser treatment [25,26].

The aim of this review was to evaluate the effects of combination treatment of PDR on regression of retinal NV, including functional outcomes and incidence of complications.

## 2. Materials and Methods

The MEDLINE and PubMed databases were searched for articles evaluating regression of NV using different combinations of the following terms: “proliferative diabetic retinopathy”, “anti-VEGF”, “panretinal photocoagulation”, and “combined treatment”. The identified articles were reviewed according to their methodology.

## 3. Results

The database search yielded a total of 34 studies that included combination treatment: anti-VEGF plus PRP in the management of PDR. Among them, 22 analyzed regression of NV and were included in the review. The sample size, follow-up duration, study design, best-corrected visual acuity (BCVA) change, central subfoveal thickness (CST) change, NV regression, reported complications and adverse events, and other data presented in each article are summarized in Table 1.

In 14 articles, the authors compared the efficacy between PRP alone and PRP combined with anti-VEGF therapy, while in three articles, the authors compared the efficacy between anti-VEGF monotherapy and combination treatment. In two articles, three arms were compared: PRP, PRP combined with intravitreal anti-VEGF therapy, and intravitreal anti-VEGF therapy alone. Two articles compared two versions of combination treatments: anti-VEGF therapy with standard Early Treatment Diabetic Retinopathy Study (ETDRS) PRP and anti-VEGF therapy with modified laser treatment. In one article, only combination treatment (IVR plus PRP) results were reported. Below, we present a synthesis of the reported findings based on the analyzed aspect of combination therapy.

## 4. Summary of the Results and Discussion

### 4.1. NV Regression

In all studies included in the review, significant regression of NV was reported for combination treatment of PDR. The studies comparing PRP alone with PRP plus anti-VEGF therapy showed greater reduction of the NV area with combination therapy [27,28,30,31,32,33,35,36,37,39,40,44,47]. Conversely, in the recent retrospective study by Si et al. [48], such an advantage of combination treatment over PRP alone was not proven within the 24-month follow-up. Regarding the strength and durability of combination treatment in terms of the treatment schedule and duration of follow-up, early studies [27,28,29,30] utilized only a few months of follow-up and one or two intravitreal injections. Despite prominent NV regression in the first month’s post-treatment, the effect diminished over time without further management [28,29]. Notably, it is essential to review subsequent studies with developed retreatment criteria and long-term follow-up. Figueira et al. [39] conducted a randomized prospective multicenter research (PROTEUS study) comparing PRP plus intravitreal ranibizumab (IVR) with PRP alone within 12 months of follow-up. The study design involved three monthly injections of IVR and allowed repetition of PRP treatment in case of persistent NV. The reduction of the total NV area was significantly greater in the PRP plus group; however, complete NV regression was obtained only in 43.9% of cases in that group. Moreover, the advantage of combination treatment was significant for neovascularization elsewhere (NVE) but not for neovascularization at disc (NVD). The rate of NV relapse after initial improvement was similar in both groups, but NV reoccurred after complete regression only in the combination group. NVD resistance to treatment was also observed by Zhou et al. [36]. Combination treatment yielded complete regression of NVE in 100% of cases in the combination group, but complete regression was noted only in 50% of the eyes.

Generally, despite the superiority of combination treatment to PRP in eliciting reduction of NV, total regression of NV was not achieved in 100% of the eyes in only one study. As an adjunct, anti-VEGF therapy improves morphological outcomes but does not provide complete cure.

PRP combined with anti-VEGF therapy and anti-VEGF monotherapy were rarely analyzed among the studies reviewed [38,43]. In the study by Messias et al. [38], regression of NV was similar between the IVR monotherapy group and two laser plus IVR groups. However, notably, repeated injections were allowed in cases of incomplete NV regression or DME; thus, the study protocol included no limit for the number of IVR sessions during the study. The study by Chatziralli et al. [43] utilized a similar design and demonstrated more optimistic results. The authors found total NV regression in the combination group (IVR plus PRP) at the end of the 24-month follow-up. Nevertheless, during that period, patients received a mean number of 11 intravitreal injections in the combination group compared with 14 intravitreal injections in the IVR monotherapy group. Hence, it is plausible that discontinuation of intravitreal treatment can result in NV relapse in some cases.

PRP alone, PRP plus anti-VEGF therapy, and anti-VEGF therapy alone were compared in three studies [34,41,46]. Figueira et al. [34] did not report any significant difference in the total NV regression between the groups, although the PRP alone group showed poorer NVD regression than did the IVR and IVR plus groups. Notably, the proportion of patients with total NV regression at the end of the study was small in all groups (<50%). Lang et al. [41] reported similar results during 12 months of follow-up in their PRIDE study. Despite significant reduction of the NV area in the IVR and IVR plus groups, complete NV regression at 12 months was noted in a minority of patients in each group. The same authors reported the results of the second year of observation of patients from their PRIDE study, during which patients were treated under real-life conditions, and anti-VEGF medications were rarely administered [49]. Most patients received supplementary PRP treatment. As a result, the NV area significantly increased, prompting the authors to conclude that discontinuation of anti-VEGF treatment for PDR might result in increased NV area and visual loss.

Shahraki et al. [46] also reported superiority of combination treatment (IVB plus modern PRP) over IVB alone and PRP alone in reducing the leakage area. The difference was prominent in the pairwise comparison between combination treatment and PRP alone (*p* = 0.003). The study design allowed rescue IVB sessions in cases without NV regression or in cases of DME. Leakage from NV was reduced but was still noted at the endpoint of the study. These findings indicate that complete NV regression should not be expected after only a limited number of intravitreal injections.

The decision to use combination treatment instead of PRP alone or anti-VEGF monotherapy for PDR must be evaluated in the context of the results of the Protocol S study by the DRCR net group [50]. This milestone study favored IVR over PRP in terms of functional outcomes: patients treated with intravitreal injections had a better visual field and lower incidence of DME. However, the long-term observation of patients showed that intravitreal therapy alone for PDR was burdened with a high proportion of patients who were lost to follow-up and eventually developed serious complications, such as retinal detachment and iris NV, which were less frequent in patients treated with PRP alone [24]. Hence, patient compliance is a crucial factor that determines the long-term outcomes of patients treated with anti-VEGF therapy alone [51,52]. Moreover, the cost utility of intravitreal monotherapy for PDR is less favorable than that of PRP. The favorable results of combination treatment for PDR suggest its potential superiority over IVR alone and PRP alone.

### 4.2. Number of Required Laser Spots/Energy

In most studies reviewed, the standard ETDRS protocol for PRP was applied, but modified protocols were also utilized and evaluated [38,42,45,46]. Two of these studies compared the effects between PASCAL laser combined with anti-VEGF therapy and classic ETDRS laser combined with the same anti-VEGF therapy [38,42]. Both lasers proved to be equally effective in reducing NV leakage. Shahraki et al. [46] evaluated a modified PRP protocol that involved PRP anterior to the equator only in combination with IVB. This treatment proved to be superior to PRP alone and IVB alone in causing regression of NV at the end of 12-month follow-up. Toscano et al. [45] compared classic PRP plus IVR with PRP at ischemic areas only plus IVR and found no difference in the outcome between both treatments.

Messias et al. [33], Figueira et al. [39], Yan et al. [35], and Sun and Qi [47] reported that combination treatment reduced the number of laser spots or energy compared with PRP alone.

As shown in the analyses of the results of the reviewed studies, retina-sparing techniques, including reduction of the number of ETDRS PRP spots, multispot PASCAL laser (number of produced spots is generally smaller), PRP at ischemic areas only, and PRP limited to the area anterior to the equator in combination with anti-VEGF therapy were equally as effective as protocols involving a traditionally larger retinal ablation. To date, the number of available studies on the subject is limited, indicating the need for further research. Nevertheless, existing evidence shows that intravitreal therapy as an adjunct to peripheral laser might save some function of the retina owing to a smaller number of required laser spots.

### 4.3. Treatment Schedule

The analyses of the treatment schedules applied in the studies reviewed showed a variety of approaches used and a lack of precise recommendations for combination treatment of PDR. The studies with short follow-up (1–6 months) applied a limited number (one to two) of intravitreal injections in combination with PRP at the beginning of the study and sometimes after the completion of PRP. Such approach was burdened with the loss of the effect of treatment and relapse of NV, as mentioned earlier [28,29]. The long-term follow-up studies used more than one or two intravitreal injections in cases without NV regression or in cases of NV relapse throughout the study. In all these studies, a necessity for the continuation of intravitreal treatment after the application of initial doses was indicated [41,42,43,45,46,47,48]. These findings indicate that total regression of NV cannot be obtained with only a small number of anti-VEGF injections. The number of required intravitreal treatments is similar for combination therapy and intravitreal therapy alone [34,38,41]. Only one study reported fewer injections for combination therapy [45].

### 4.4. Adverse Events: VH and Need for PPV

VH and subsequent treatment with PPV during follow-up were not frequently reported in any study. Moreover, the incidence of these adverse events was not analyzed in all studies. Nevertheless, the incidence of these complications in most studies was higher in the PRP alone group than in the PRP plus group [30,31,34,36,39,46,47]. For example, Figueira et al. [34] reported an incidence of 30.8% and 9% in these groups, respectively, while Sun and Qi [47] noted these complications in 27 and seven eyes in their sample of 165 eyes, respectively. These findings are consistent with those of the Protocol S study, which compared anti-VEGF treatment with PRP for PDR. In the Protocol S study, the need for PPV was higher in the PRP group: 15% vs. 4% [50]. Nevertheless, the protective effect of anti-VEGF treatment depends on regular application of intravitreal injections.

### 4.5. BCVA Change

The BCVA is expected to improve in patients with PDR with concomitant DME. The reviewed studies often presented results for the whole cohort of patients with PDR, without division to those with and without DME, making it difficult to analyze the real impact of treatment modalities on the BCVA. Accordingly, it is essential to review studies excluding patients with PDR with concomitant DME from the analysis. Among the analyzed studies, four specifically included patients with PDR but without DME at baseline [27,30,39,41]. In two of these studies [27,30], no significant difference was noted in the final BCVA between the PRP alone group and PRP combined with anti-VEGF therapy group. No improvement of vision was also observed. In the PROTEUS study by Figueira et al. [39], the BCVA was better in the IVR plus group than in the PRP alone group only after correction for age. Nevertheless, the mean BCVA slightly declined in both groups. The PRIDE study by Lang et al. [41] revealed only a minor improvement in the BCVA of the IVR alone group and a decline in the BCVA of the PRP plus and PRP alone groups. A significant difference was noted only between the IVR and PRP alone groups. These findings indicate a tendency for sustainability of the BCVA with anti-VEGF therapy as an adjunct to PRP. PRP alone is usually associated with BCVA decline.

The composition of the study groups that included patients with DME is similar between the different arms of the reviewed studies (the same proportion of patients with DME). Thus, some conclusions can be drawn. The studies that included patients with DME generally showed either improvements in the BCVA [28,29,32,35,37,38,40,42,43,44,47] or stabilization [31,41,45] with combination treatment and, consequently, decline or rare stabilization with PRP alone. In the long-term, the effect was transient when anti-VEGF therapy was applied only at the beginning of the study and not repeated afterward [36]. Only few studies showed no variations in the BCVA between the analyzed groups [33,34].

### 4.6. CST Reduction

The studies that included patients without DME generally did not show a significant impact of anti-VEGF therapy as an adjunct to PRP on the CST [27,30,39,41]. In the PRIDE study by Lang et al. [41] and subgroup analysis by Shahraki et al. [46], anti-VEGF therapy alone yielded the lowest CST values at the end of follow-up.

Patients with DME presented better morphological responses to combination treatment with anti-VEGF therapy [29,31,34,35,36,38,39,44,47] than to PRP alone at least at some point throughout the studies.

### 4.7. Other Data

The other data reported in the analyzed studies showed that PRP yielded more functional retinal damage than did combination therapy or intravitreal therapy alone. Reductions of ERG readings [33,38] as well as deficits in the visual field [46] were greater in the PRP alone group. One study reported improvements in the condition of the ellipsoid zone and external limiting membrane among the PRP plus IVR and IVR monotherapy groups [43]. The recent study by Si et al. [48] evaluated changes in angio-OCT vascular parameters after PRP plus and PRP treatments. The foveal avascular zone area and number of microaneurysms decreased more, and the vessel density of the superficial capillary plexus significantly increased in the combination group. In the study by Zhou et al. [36], the times to resorption of VH and regression of NV were significantly shorter with anti-VEGF therapy as an adjunct to PRP (resorption of VH: 12.1 weeks for PRP alone vs. 8.4 for combination treatment, regression of NVD: 15 weeks for PRP alone vs. 12.5 weeks for combination treatment). All these data prove the additional benefits of anti-VEGF injections as an adjunct therapy for PDR.

## 5. Conclusions

This review demonstrates that PRP combined with anti-VEGF therapy is superior to PRP alone in the management of PDR. This combination treatment yields better and faster NV regression and a lower incidence of serious complications requiring PPV. Nevertheless, complete NV regression is not achieved with any treatment in 100% of patients. Moreover, further research is needed to establish the most effective schedule for intravitreal injections as an adjunct to PRP. The current literature shows that in some cases, cessation of anti-VEGF injection in combination treatment for PDR can lead to relapse of NV.

## Figures and Tables

**Table 1 diagnostics-14-00031-t001:** Summary of the findings of the included studies on combination treatment of PDR.

	Study	No. of Eyes Included	Follow-up Duration (Month)	Study Design	Primary Study Outcome	BCVA Change	CST Change	NV Regression	Incidence of VH and Need for PPV	Other Data
1.	Tonello et al., 2008 (IBeHi study) [27]	30 eyes without DME	4	Prospective; patients with high-risk PDR; PRP group (15 eyes) vs. PRP plus IVB group (15 eyes); PRP performed at 1 and 3 weeks and IVB at 3 weeks	BCVA and regression of the NV area assessed on FA at baseline and 4, 9, and 16 weeks	No significant difference between the groups; BCVA change from 0.26 logMAR at baseline in both groups to 0.31 logMAR in the PRP alone group and 0.29 logMAR in the PRP plus group	CST not observed	Significant reduction in the NV area in the PRP plus group at each timepoint compared with the baseline (11.15 mm^2^ at baseline vs. 4.46 mm^2^ at 16 weeks); increase in the NV area at 16 weeks compared with that at 1 and 3 weeks in the PRP plus group; no change in the NV area in the PRP alone group at any timepoint	Not observed	
2.	Mirshahi et al., 2008 [28]	80 eyes, excluding those with DME	4	Prospective; 40 patients with bilateral HR-PDR treated with PRP alone (one eye) vs. PRP combined with IVB 1.25 mg (fellow eye) at the first session	Regression of NV on FA and factors influencing the recurrence of NV	Data reported for combined treatment: BCVA change from 1.21 logMAR to 0.7 logMAR (data without division into PRP and PRP-naive subgroups or DME present/absent subgroup); *p* < 0.00001	Not reported	FA performed at 6 and 16 weeks post-treatment; at 6 weeks, complete regression of NV noted in 87.5% of patients in the IVB plus PRP group and 25% in the PRP alone group (*p* < 0.005); at 16 weeks, complete regression noted in 25% in both groups	Not reported	Recurrence of NV strongly correlated with higher HbA1c levels (*p* = 0.004)
3.	Arevalo et al., 2009 [29]	44 eyes (18 eyes with DME)	6	Retrospective; study of the effect of at least one injection of bevacizumab 1.25 or 2.5 mg in patients with PDR; 77% of patients previously treated with PRP but at least 6 months before IVI; 23% treatment-naive	Regression of NV on FA, BCVA, and CST	BCVA change from 1.21 logMAR to 0.7 logMAR (data without division into PRP and PRP-naive subgroups or DME present/absent subgroup); *p* < 0.00001	For patients with DME: mean CST reduction from 487.4 µm to 260.6 µm (*p* < 0.00001)	Complete regression of NV (lack of leakage on FA) noted in 61.4% of patients, partial regression in 34.1%, and no regression in 4.5%; second injection required in 47.7% of patients	VH occurrence: 2.2% (one patient); need for PPV: 2.2% (one patient with TRD)	No systemic AE
4.	Shin et al., 2009 [30]	24 eyes without DME	1.5	Retrospective; patients with high-risk PDR; one eye of the same patient treated with PRP alone (12 eyes) vs. other eye of the same patient treated with PRP plus single IVB (12 eyes)	BCVA, IOP, and NV area regression at 6 weeks on FA	No significant BCVA improvement in any group; BCVA after treatment: 0.28 logMAR in the PRP plus group vs. 0.24 logMAR in the PRP alone group (*p* = 0.916)	CST not observed	Significant NV reduction in both groups; greater reduction in the PRP plus group than in the PRP alone group: 12.44 pixels × 10^3^ vs. 6.48 pixels × 10^3^ (*p* = 0.038)	VH occurrence: two patients in the PRP only group; need for PPV: two patients with VH	IOP change: no significant increase and no difference between the groups
5.	Filho et al., 2011 [31]	29 eyes, including those with DME	12	14 eyes treated with PRP alone vs. 15 eyes treated with PRP combined with IVR (retreatment at 16 and 32 weeks when needed); patients with DME included and treated with focal laser	Total area of fluorescein leakage on FA, BCVA, and CST at 16, 32, and 48 weeks	Decrease in the BCVA by 0.08 logMAR in the PRP alone group; no change in the PRP plus group; significantly better BCVA at endpoint in the between-group comparison	Significant increase in the CST in the PRP alone group (+18.1 µm) (*p* < 0.05) and tendency to decrease in the PRP plus group; between-group comparison data not significant (*p* = 0.7106)	Significantly smaller NV area on FA at 48 weeks in the PRP plus group than in the PRP alone group (*p* = 0.029)	VH occurrence: one patient in the PRP only group; need for PPV: not reported, but one case of TRD occurring in the PRP only group	
6.	Ahmad and Jan, 2012 [32]	54 eyes (25 eyes with DME)	3	Prospective; 27 eyes with PDR treated with PRP vs. 27 eyes treated with PRP plus single IVB post-second PRP; 12 patients with DME (PRP alone group) and 13 patients with DME (PRP plus group)	Change in the area of NVD (% of DA) and NVE (DD)	Worsening of the BCVA from 0.3 to 0.4 logMAR in the PRP alone group and improvement from 0.3 to 0.1 logMAR in the PRP plus group (*p* = 0.00002)	Not reported	PRP alone group: no significant change in NVD (40%) and NVE (2 DD) PRP plus group: significant decrease in NVD from 40% to 11% (*p* = 00004) and NVE from 2 DD to 0.75 DD (*p* = 00008)	Not reported	
7.	Messias et al., 2012 [33]	20 eyes, excluding those with DME	12	Prospective; patients with HR-PDR treated with PRP alone (*n* = 9) vs. PRP plus IVR at the first PRP session (*n* = 11); evaluation of the BCVA and NV on FA conducted at baseline and 16, 32, and 48 weeks; ERG performed at baseline and 48 weeks; retreatment: additional laser or additional IVR allowed at 16 and 32 weeks in case of active NV on FA	ERG, BCVA, and leakage area on FA	No significant difference between the groups; no significant change in the BCVA throughout the study; BCVA after treatment: 0.37 logMAR in the PRP alone group vs. 0.28 logMAR in the PRP plus group (*p* > 0.05)	Not reported	Significant reduction in leakage on FA in both groups (*p* < 0.05); significantly larger reduction in the PRP plus group (*p* = 0.0074)	Not reported	ERG: ROD b-wave amplitude—reduction significantly larger in the PRP group (*p* = 0.024); combined response b-wave also as above (*p* = 0.0094); number of required laser spots larger in the PRP alone group than in the PRP plus group (2736 vs. 1636)
8.	Figueira et al., 2016 [34]	32 eyes, excluding those with DME	12	RCT, prospective; patients with HR-PDR; three arms: IVR alone (9 eyes), IVR plus PRP (10 eyes), and PRP alone (13 eyes); patients treated with IVR receiving three loading injections with monthly intervals; additional treatments allowed in case of persisting or recurring NV; evaluation conducted at baseline and 3, 6, and 12 months	BCVA, CST, and NV area	No significant difference between the groups throughout the study; no significant variations in the BCVA in any arm of the study during follow-up	No significant differences between the groups during the study, except at 3 months for the IVR plus PRP group when the CST significantly decreased	Complete NVE regression at 12 months: 44.4% in the PRP plus IVR group, 37.5% in the IVR group, and 30.8% in the PRP alone group (no significant difference); complete NVD regression at 12 months: 37.5% in the PRP plus IVR group, 40% in the IVR alone group, and 22.2% in the PRP alone group (significant difference in favor of the PRP plus IVR or IVR alone group)	Rate of complications such as VH or need for PPV significantly higher in the PRP alone group than in the IVR alone and PRP plus groups (30.8% vs. 11.1% and 9%)	
9.	Yan et al., 2016 [35]	83 eyes with DME and severe NPDR or PDR	6	Prospective; 37 eyes receiving PRP plus single IVR prior to laser vs. 38 eyes treated with PRP alone	Change in the BCVA, macular leakage area, and CST	BCVA similar between baseline and 1 month; significant improvement in the PRP plus group at 3 and 6 months	CST significantly lower in the PRP plus group at 1, 3, and 6 months	Leakage area significantly smaller in the PRP plus group at 1 and 3 months; no significant difference at 6 months	Not observed	IOP: no difference; required laser energy significantly lower in the PRP plus group
10.	Zhou et al., 2016 [36]	36 eyes with HR-PDR, including 18 eyes with DME	6	Retrospective; PRP plus single IVB (18 eyes) vs. PRP alone (18 eyes); PRP performed at 1 and 3 weeks; evaluation conducted at 12, 16, and 24 weeks	Change in the BCVA, NV leakage area on FA, and CST	Better BCVA improvement in the PRP plus group than in the PRP alone group at 24 weeks but not at 12 and 16 weeks (by 0.1 logMAR vs. 0.03 logMAR)	Increase in the PRP group, significant decrease in the PRP plus group, but no significant difference between the groups	Significantly greater reduction of the NV leakage area in the PRP plus group at all timepoints (by 6.4 vs. 3.4 mm^2^ at 24 weeks). At 1 year, complete regression of NVE noted in 100% of the eyes in the PRP plus group vs. 83.3% in the PRP alone group and NVD in 50% of the eyes in the PRP plus group vs. 33% in the PRP alone group	VH occurrence: two patients in the PRP only group; need for PPV: two patients with VH	Vitreous clear-up time for cases complicated by VH: 12.1 weeks in the PRP plus group vs. 8.4 weeks in the PRP alone group; NVD regression time: 15 weeks vs. 12.5 weeks in favor of the PRP plus group
11.	Ali et al., 2018 [37]	60 eyes with PDR, including those with DME	1	Prospective, randomized; PRP plus single IVB prior to laser (30 eyes) vs. PRP alone (30 eyes); analysis of two age groups (40–52 years and 53–65 years)	Change in the BCVA and regression of NVD and NVE	Older group with a poorer BCVA at baseline; significantly better BCVA improvement in the PRP plus group (from 0.64 logMAR to 0.49 logMAR vs. no improvement in the PRP only group)	Not analyzed	Significant NV regression in both groups (*p* < 0.001); regression in the PRP plus group significantly greater: NVD/DD%: 31.27 to 11.40 and NVE/DD: 3.30 to 1.50 vs. PRP only group: NVD/DD%: 31.13 to 29.53 and NVE/DD: 3.33 to 3.17 (*p* < 0.001)	Not analyzed	No difference in the IOP between the groups
12.	Messias et al., 2018 [38]	43 eyes, including those with DME	12	Prospective; IVR plus PASCAL laser (15 eyes) or IVR plus ETDRS laser (15 eyes) vs. IVR alone (13 eyes); IVR at baseline and repeated when required for each group (monthly for DME and every 12 weeks for NV); PRP performed in one session for PASCAL and two sessions for ETDRS laser	Change in the BCVA, leakage area/% on FA, CST, and ERG reading; evaluation conducted every month and at 12, 24, and 48 weeks for ERG	No significant difference between the groups (*p* > 0.05); significant improvement in each group (by 0.1–0.3 logMAR)	No significant difference between the groups; small significant reduction in the CST in both PRP plus groups	Significant leakage reduction on FA in all groups (55.9% of cases in the ETDRS plus group, 73.1% in the IVR group, and 73.3% in the PASCAL plus group); no significant difference between the groups; stabilization of regression starting at 24 weeks	Not reported	No significant difference in the number of IVR injections (4.2 in the ETDRS group, 5.5 in the PASCAL group, and 4.6 in the IVR only group); significant a- and b-wave amplitude reduction for dark- and light-adapted ERG in the ETDRS and PASCAL plus groups; only minor dark-adapted b-wave reduction in the IVR group
13.	Figueira et al., 2018 (PROTEUS study) [39]	77 eyes with HR-PDR without DME	12	Prospective, randomized, multicenter; IVR (three monthly injections) plus PRP (41 eyes) vs. PRP alone (46 eyes); possible repetition of PRP treatments	Regression of the total NV (NVD plus NVE) on FA, BCVA, CST, and time to total NV regression; evaluation conducted every month	BCVA change in the IVR plus group (−0.9 letters vs. −5.8 letters) vs. PRP alone group; no significant difference in the final BCVA between the groups (*p* = 0.104); after age correction, BCVA significantly higher in the IVR plus group (*p* = 0.031)	Significant difference between the groups at 3 and 7 months (CST lower in the PRP plus group); no significant difference at 12 months (*p* = 0.078); treatment for DME needed in two cases in the PRP alone group	Total NV reduction in 92.7% of patients in the PRP plus group vs. 70.5% in the PRP alone group (*p* = 0.009); complete NV total regression: 43.9% in the PRP plus group vs. 25.0% in the PRP alone group (*p* = 0.066); regression rate higher for NVE in the PRP plus group but not for NVD; NV regression significantly earlier in the combination group: 3.6 vs. 7.0 months; recurrence of NV after total regression only in the PRP plus group (66.6%)	VH occurrence: 20 patients (nine in the PRP group and 11 in the combination group); need for PPV: six patients (five in the PRP group and one in the combination group)	Mean number of PRP treatments and laser spots larger in the PRP monotherapy group; seven patients discontinued the trial owing to DR complications in the PRP alone group and one patient in the combination group; difference significant at *p* = 0.04
14.	He et al., 2020 [40]	15 patients, 30 eyes with treatment-naive HR-PDR, six eyes with DME (three in each group)	6	IVC plus PRP (15 eyes) vs. PRP alone (15 other eyes); two IVC injections—one before and one after PRP	NV regression on FA, BCVA, CST, FAZ area, and vessel density (superficial plexus); evaluation conducted at baseline and 3 and 6 months	BCVA improvement in the PRP plus group and reduction in the PRP alone group; significant difference noted	No significant difference between the groups at any timepoint	Significantly greater NV leakage reduction in the PRP plus group than in the PRP alone group at 3 and 6 months (*p* < 0.0001) (−7.61 vs. 3.24 mm^2^ and −11.1 vs. 6.10 mm^2^); complete regression at 6 months in 13.3% of patients in both groups	Not reported	No significant difference in the FAZ size and vascular density between the groups at any timepoint
15.	Lang et al., 2020 (PRIDE study) [41]	106 eyes with PDR without DME; 83 eyes completing the full study protocol	12	Prospective; efficacy of IVR alone (36 eyes) vs. PRP alone (36 eyes) vs. IVR plus PRP (36 eyes); IVR administered as a loading dose of three monthly injections and PRN thereafter; rescue laser permitted	Change in the area of NV; analysis available in 99 eyes	Minor BCVA improvement in the IVR group and decline in the two other groups; significant difference between the IVR and PRP alone groups (5.5 letters)	Decrease in the CST in the IVR group and increase in the PRP and PRP plus groups; significant difference between the IVR and PRP groups (*p* = 0.0003) and between the IVR and combination groups (*p* = 0.0357), both in favor of the IVR group	Decrease in the NV area from 9.39 to 2.7 mm^2^ in the IVR group, from 5.4 to 4.08 mm^2^ in the PRP alone group, and from 4.08 to 1.96 mm^2^ in the IVR plus group; significant difference between the IVR and PRP groups but not between other groups; complete regression of NV at 12 months observed in 27.6% of patients in the IVR group, 7.7% in the PRP group, and 17.9% in the combination group	VH occurrence: 18 patients; need for PPV: five patients (two in the PRP only group and three in the combination group)	No significant change in the ETDRS severity scale score between baseline and 12 months
16.	Barroso et al., 2020 [42]	40 eyes, including 12 eyes with DME	12	Prospective, randomized; PRP ETDRS laser plus single IVR vs. PRP PASCAL plus single IVR vs. IVR alone; IVR administered 180 min after PRP; grid laser applied to patients with DME; retreatment with IVR in case of active NV leakage and/or presence of DME	Change in the BCVA, CST, and NV leakage on FA (mm^2^)	No significant difference between the groups at endpoint follow-up; significant improvement at endpoint in all groups	No significant difference between the groups at endpoint follow-up; significant reduction at endpoint in the PASCAL and IVR alone groups; no significant reduction in the ETDRS group	No significant difference between the groups; significant leakage reduction on FA at all follow-up visits in the intragroup analysis	Not reported; lost to follow-up patients: need for PPV in patients (one with persistent VH and one with TRD)	No significant difference in the number of IVR injections between the groups; number larger in patients with DME
17.	Chatziralli et al., 2020 [43]	47 eyes with PDR and DME	24	Prospective; PRP plus IVR (23 eyes) vs. IVR alone (24 eyes); at least three sessions of IVR in the loading phase in both groups and PRN thereafter; retreatment in case of persistent NV and/or DME	Change in the BCVA, CST, and NV leakage on FA	No significant difference (*p* = 0.064) between the groups at endpoint; the combination group gaining +17.1 ETDRS letters; the IVR group gaining +12.3 letters compared with the baseline	No significant difference at endpoint comparison; the IVR alone group gaining significantly higher reduction at 12 months; both groups achieving significant reduction from baseline (*p* < 0.001)	Significant NV regression in both groups; significantly greater regression of NVD in the combination group (NVD present in 0% of patients in the combination group vs. 17.4% in the IVR alone group at the final visit) (*p* = 0.049); no difference regarding regression of NVE (0% vs. 4.3% at the final visit) (*p* = 489)	VH occurrence: five patients (four in the monotherapy group and one in the combination group); need for PPV: five patients with VH	Improvement in the EZ and ELM conditions in both groups without a significant difference; number of injections larger in the monotherapy group (14 vs. 11)
18.	Rebecca et al., 2021 [44]	76 eyes with HR-PDR, including 34 eyes with DME	6	Prospective, randomized; PRP plus IVB (38 eyes) vs. PRP alone (38 eyes); IVB administered before and after PRP (two injections)	NV regression; NVD expressed in DD%, NVE in DD size, BCVA, and CST	BCVA reduction in the PRP group and improvement in the combination group (endpoint values: PRP group: 0.42 logMAR, combination group: 0.1 logMAR; *p* = 0.00002)	CST at endpoint: PRP alone group: 299 μm, combination group: 240 μm; significant difference at *p* = 0.000011	Residual NVD: 40% DD in the PRP group (no change from baseline) vs. 12% DD in the combination group (significant reduction) (*p* = 0.00004); residual NVE: 2.1 DD in the PRP group (no significant change) vs. 0.7 DD in the combination group (reduction) (*p* = 0.0001)	Not reported	
19.	Toscano et al., 2021 [45]	28 eyes, including those with DME	12	RCT, prospective; PRP plus IVR vs. IVR plus PRI (laser at ischemic areas only); IVR administered after the first session and quarterly thereafter in case of active NV or DME	BCVA, CST, NV leakage on FA every 12 weeks, and ERG reading at baseline and 3 months	No significant difference between the groups and throughout the follow-up in each group	No significant difference between the groups and throughout follow-up in each group	Significant leakage reduction on FA from 4 weeks onward; no significant difference between the groups (PRP group: reduction from 16.1 to 2.89 mm^2^ vs. PRI group: from 9.97 to 0.67 mm^2^)		ERG, number of IVI: no significant difference between the groups
20.	Shahraki et al., 2022 [46]	153 eyes, including 91 eyes with DME	12	Prospective, randomized; IVB vs. PRP vs. modified laser therapy plus IVB PRP group: 51 eyes; 1200–2400 laser spots applied in 3 months; rescue IVB possible in case of VH or progression of NV; in case of NV improvement at 4 and 8 months, no further intervention considered IVB group: 52 eyes; four monthly IVB sessions; in case of NV worsening or VH development, four IVB sessions added; in case of NV persistence, two IVB monthly or bimonthly sessions added; in case of NV improvement, no IVB performed; same strategy applied at 8 months; in case of NV worsening, rescue laser considered an option plus four monthly IVB sessions Modern combined treatment group: 50 eyes; two monthly IVB sessions followed by PRP (400–800 spots anterior to the equator); in case of NV progression or VH development, four IVB sessions added; in case of NV worsening, rescue laser performed at 8 months, with the rescue laser rate higher in the IVB group (*p* = 0.001) Evaluation conducted at baseline and 4, 8, and 12 months	BCVA, NV regression on FA, CST, and MD-VF	No significant difference in the final BCVA between the groups at the final follow-up (*p* = 0.77)	No significant difference between the groups at the final follow-up (*p* = 0.24); the PRP group having the highest rate of new-onset DME, followed by the IVB group and modified combination group; among the non-DME subgroups, the IVB group having the lowest final CST, followed by the modified combination group and the PRP group (*p* = 0.03)	Leakage area difference significant at 8 and 12 months (*p* = 0.03 and *p* = 0.001, respectively); lowest decrease in NV in the IVB group, followed by that in the PRP group and modified combination group; smaller final leakage area in the modified combination group than in the PRP group (*p* = 0.003) and IVB group (*p* = 0.001); PRP group vs. IVB group, in favor of the PRP group (*p* = 0.009); similar results regarding the number of NVE at the final follow-up, with the lowest number noted in the modified combination group, followed by that in the PRP group and IVB group; combined protocol as effective as full PRP and more effective than IVB in reducing NV and leakage	VH occurrence: 28 patients (10, 11, and 7 in the PRP, IVB, and combination groups, respectively); need for PPV: four patients (one, three, and zero in the PRP, IVB, and combination groups, respectively); lowest rate of retinal detachment requiring vitrectomy, neovascular glaucoma, iris NV, new vitreous hemorrhage noted in the modified combination group (*p* = 0.032)	Lowest MD-VF in the PRP group (*p* < 0.05); number of IVB injections: 3.5 in the PRP group, 7.4 in the IVB group, and 6.2 in the combination group; number of visits among the patients with DME larger in the IVB group; number of injections needed for treating DME within 1 year similar between all groups (*p* = 0.112)
21.	Sun and Qi, 2023 [47]	165 eyes, including 76 eyes with DME	12	Retrospective; IVC plus PRP (79 eyes) vs. PRP alone (86 eyes); combination group initially treated with three IVC sessions monthly; continuation in case of NV persistence at 3 months; for the PRP monotherapy group, retreatment with PRP at 3 months in case of non-regression of NV; evaluation conducted at baseline and 3, 6, 9, and 12 months	NV regression rate, BCVA, and CST	Average BCVA significantly greater in the combination group (*p* < 0.05) at each visit; proportion of patients with improved, unchanged, and decreased BCVA: 86.08%, 11.39%, and 2.53% in the IVC plus PRP group and 23.26%, 55.81%, and 20.93% in the PRP monotherapy group, respectively	Average CST significantly lower in the combination group (*p* < 0.05) at 3, 6, 9, and 12 months; proportion of patients with decreased, unchanged, and increased CST: 74.68%, 25.32%, and 0% in the IVC plus PRP group and 30.23%, 39.54%, and 30.23% in the PRP monotherapy group, respectively	NV regression rate significantly higher in the combination group (*p* < 0.001); proportion of patients with complete NV regression, partial NV regression, and no NV regression or increase of sustained NV: 70.88%, 29.12%, and 0.0% in the IVC plus PRP group and 15.12%, 58.14%, and 26.74% in the PRP group, respectively	VH occurrence: 34 patients (27 in the PRP group and seven in the combination group); need for PPV significantly lower in the IVC plus PRP group (seven eyes; 8.86%) than in the PRP alone group (27 eyes, 31.40%) (*p* < 0.001)	Smaller number of laser spots in the combination group: 1453 ± 87 vs. 2267 ± 94 (*p* < 0.05); no significant difference in the total number of laser treatments between the groups
22.	Si et al., 2023 [48]	82 eyes with HR-PDR, including those with DME	24	Retrospective; IVC plus PRP (50 eyes) vs. PRP alone (32 eyes); combination group treated with PRP and three IVC sessions monthly and IVC thereafter in case of NV persistence or recurrence; in the PRP monotherapy group, additional laser applied in case of non-regression of NV; PRP baseline regimen: four sessions, each with 300–400 spots; evaluation conducted at baseline and 6, 12, 18, and 24 months	Retinal vein changes—significant vein diameter regression in favor of the combination group (*p* < 0.01) at 12, 18, and 24 months; BCVA, CST, NV, and superficial capillary plexus	Between-group comparison results not provided; the combination group pre- vs. post- treatment measurement at each evaluation point presenting with significant improvements (*p* < 0.01); significant improvements in the BCVA in the first 6 months; therapeutic effect maintained throughout the study	Between-group comparison not performed; combination group pre- vs. post- treatment measurement at each evaluation point (*p* < 0.01); significant CST reduction in the first 6 months; therapeutic effect maintained throughout the study	No significant difference between the two groups at endpoint; significant decrease in the NV area in the first 6 months in both groups but stability during follow-up; significant decrease in each group at each follow-up point	Not reported	Decrease in the FAZ area in the combination group; significant decrease in the number of microaneurysms in favor of the combination group; significant reduction in the hard exudate area in each group, without significant difference between the groups; significant increase in the superficial capillary plexus vessel density in the combination group

BCVA—best-corrected visual acuity, CST—central subfoveal thickness, PRP—panretinal photocoagulation, PRI—panretinal photocoagulation at ischemic areas only, ETDRS—Early Treatment Diabetic Retinopathy Study, NV—neovascularization, NVD—neovascularization at disc, NVE—neovascularization elsewhere, DME—diabetic macular edema, FA—fluorescein angiography, VH—vitreous hemorrhage, PDR—proliferative diabetic retinopathy, HR-PDR—high-risk proliferative diabetic retinopathy, PPV—pars plana vitrectomy, DA—disc area, DD—disc diameter, IVB—intravitreal bevacizumab, IVI—intravitreal injection, IVR—intravitreal ranibizumab, IVC—intravitreal conbercept, IOP—intraocular pressure, RCT—randomized controlled trial, PRN—pro re nata (as needed), FAZ—foveal avascular zone, EZ—ellipsoid zone, ELM—external limiting membrane, TRD—tractional retinal detachment, AE—adverse effect, MD-VF—mean deviation of visual field, ERG—electroretinogram, HbA1c—glycated hemoglobin.

## Data Availability

Not applicable.

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
