# Peer review of "Regression of Neovascularization after Panretinal Photocoagulation Combined with Anti-VEGF Injection for Proliferative Diabetic Retinopathy—A Review"

_diagnostics, 2023, doi:10.3390/diagnostics14010031_

Round 1

Reviewer 1 Report

Comments and Suggestions for Authors

The manuscript is well written and very useful for the medical retina specialists. I see minor issue:  the title is about "Regression of neovascularization", but the second paragraph in results is BCVA change and the third - CST reduction - for my opinion secondary outcomes for the topic of this manuscript. 
It seems to me that it is better to move these paragraphs after the main data of the manuscript - after "Adverse events: VH and need for PPV". But authors may decide order of this information.

References: rows 415-416 why are all words in the title of the article #46 are written by capital letters? 

Author Response

Thank you very much for all the remarks.
1.    We changed the order of the paragraphs.
2.    We performed the corrections in the references.

MG

Reviewer 2 Report

Comments and Suggestions for Authors

The author discuss the treatment for neovascularization in PDR. The authors enrolled adequate literatures with good quality. In addition, the writing and study design of this study are sound. In think ths paper can be accepted as current form.

Comments on the Quality of English Language

Minor editing of English language required

Author Response

Thank you for the kind remarks.

MG